# Functional interrogation of candidate *cis*-regulatory elements at the *LDLR* locus

Kyle Leix[1⊙], Candilianne Serrano-Zayas[1⊙], Hitarthi S. Vyas[1], Sarah E. Graham[2], Brian T. Emmer[1,3]*

**1** Department of Internal Medicine, Division of Hospital Medicine, University of Michigan Medical School, Ann Arbor Michigan, United States of America, **2** Department of Internal Medicine, Division of Cardiovascular Medicine, University of Michigan, Ann Arbor Michigan, United States of America, **3** Frankel Cardiovascular Center, University of Michigan, Ann Arbor Michigan, United States of America

⊙ These authors contributed equally to this work.
* bemmer@med.umich.edu

## Abstract

Regulation of *LDLR* gene expression plays an important role in the development of atherosclerotic diseases including heart attack and stroke. Although *LDLR* regulation by sterol response elements has been well characterized, the functional significance of other noncoding regions at the *LDLR* locus remains poorly defined. In this study, we developed and applied a high throughput CRISPR screen to test the functional importance of candidate *LDLR cis*-regulatory elements (CREs) in their native genomic context. In total, we found 25 discrete regions to exhibit a significant impact on *LDLR* expression. For one of these regions with particularly strong activity in the first intron, we validated the presence of an enhancer by confirming that its disruption reduced endogenous *LDLR* expression while its insertion upstream of a minimal promoter augmented reporter gene expression. We then applied a massively parallel reporter assay to fine map enhancer activity within this region to a 129 bp interval that is highly conserved among vertebrates, exhibits biochemical hallmarks of enhancer activity, is enriched for transcription factor binding motifs, and contains a common genetic variant (rs57217136) that has been associated with human LDL cholesterol levels by genome-wide association studies. Overall, these findings demonstrate the power of CRISPR screening to interrogate candidate CREs and clarify the functional landscape of noncoding sequences at the *LDLR* locus.

## Author summary

Expression of the *LDLR* gene influences a person's risk of cardiovascular disease due to its impact on the concentration of cholesterol-carrying lipoproteins in the bloodstream. While some of the transcription factors and their corresponding binding sites that control *LDLR* gene expression have been well characterized, several noncoding regions exhibit features predictive of gene regulatory activity

**Data availability statement:** All relevant data are within the manuscript and its Supporting information files.

**Funding:** This research was supported by the National Institutes of Health K08-HL148552 (BTE), R01-HL167733 (BTE), R01- HL171013 (BTE), the A. Alfred Taubman Medical Research Institute (BTE), the American Heart Association 24POST1198445 (HV), and the Michigan Pioneer Fellows Program (HV). The funders had no role in study design, data collection and analysis, decision to publish, or preparation of the manuscript.

**Competing interests:** I have read the journal's policy and the authors of this manuscript have the following competing interests: SEG is a current employee of Regeneron Pharmaceuticals. CSZ is a current employee of Eli Lilly and Company.

and have not been previously tested. In this study, we leveraged advances in high-throughput CRISPR technology to systematically interrogate the functional significance of these candidate regions with a library of 12,375 gRNAs. We found evidence of activity for 25 distinct CREs, including 1 in the early first intron whose functional significance we then validated and whose precise boundaries we defined using a massively parallel reporter assay. Our findings deepen our basic understanding of *LDLR* gene regulation, suggest novel targets for therapeutic development, and illustrate the potential of high throughput genomic tools to define the functional landscape of the noncoding genome.

## Introduction

The amount of LDL receptor (LDLR) expressed in the liver is a major determinant of a person's lifetime risk of atherosclerotic cardiovascular diseases (ASCVD) including heart attack and stroke. Whereas genetic deficiency of LDLR causes familial hypercholesterolemia and early onset ASCVD, genetic variants associated with increased LDLR activity are protective against ASCVD [1,2]. Likewise, multiple classes of therapeutics that augment hepatic LDLR activity are effective in preventing the development of ASCVD [3].

Pioneering studies of *LDLR* gene regulation have firmly established the importance of SREBP transcription factors and their binding to sterol response elements (SREs) in the promoter of the *LDLR* gene [4–6]. However, eukaryotic gene regulation is a complex process in which the activity of individual transcription factors may be modulated by interactions between promoters and *cis*-regulatory elements (CREs) of noncoding genomic DNA. The total number of CREs in the human genome is estimated to be on the order of hundreds of thousands to millions [7]. For *LDLR*, several lines of evidence point toward a complexity of regulation beyond SREBP binding alone. First, different genes containing SREs in their promoters vary widely in relative expression [4]. Second, other transcription factors that modify *LDLR* expression through sterol-independent mechanisms have been identified [4,8], including in CRISPR screens by us [9] and others [10,11]. Third, human genome-wide association studies have identified several noncoding common genetic variants associated with LDL cholesterol that reside near the *LDLR* locus but do not localize directly to SREs [12–14]; a targeted study of 4 of these detected a functional influence for 2 [15]. Finally, genome-wide profiling experiments have identified candidate CREs at the *LDLR* locus that do not colocalize with SREs [16–19]. Collectively, these observations support the likelihood that *LDLR* expression is influenced by currently unrecognized CREs and their associated DNA-binding proteins.

We previously reported a high-throughput CRISPR screen of the coding genome for *trans*-regulators of cellular LDL uptake [9]. The rigor of our screening pipeline was strongly supported by our clear detection of known regulators of *LDLR* expression as well as our validation and mechanistic investigation of select novel regulators [9,20,21]. In this study, we now report our adaptation of this same CRISPR screening

approach to the noncoding landscape of the *LDLR* locus, leading to our identification of 25 functional CREs and our subsequent fine mapping of a strong enhancer in the first intron of *LDLR*.

## Results

### Design and synthesis of a custom CRIPSR library targeting putative CREs at the *LDLR* locus

To nominate candidate CREs for functional testing, we identified target regions based on multiple data sets. We first analyzed associations of common genetic variants with circulating LDL cholesterol levels in a multi-ancestry GWAS of over 1.6 million individuals [14], finding a cluster of 72 strongly associated variants that colocalize within a ~ 30 kb region around the *LDLR* transcription start site (TSS) and exhibit a high degree of linkage disequilibrium with each other; we included this entire interval in our target set (S1 Table and S1A Fig). Additionally, we performed a conditional analysis on the lead SNP in this region (rs73015024) to identify another 29 variants within 500 kb of the *LDLR* TSS that were independently associated with LDL cholesterol in individuals of European ancestry at genome-wide significance; we included 1 kb intervals centered on each of these 29 SNPs (S2 Table and S1B Fig). We also analyzed publicly available data sets of human liver tissue and liver-derived cell lines [17–19,22] to identify all regions within 100 kb of the *LDLR* TSS that exhibited biochemical features predictive of *cis*-regulatory activity, including an open chromatin state (ATAC-seq and DHS-seq peaks) and/or proximity to histone marks associated with enhancer activity (H3K27ac ChIP-seq peaks). In this same interval we also identified regions with a high degree of evolutionary conservation among primates. Finally, we included all regions in this interval that were predicted by the ENCODE project to exhibit *cis*-regulatory activity [22]. In total, we identified 12,375 gRNA targets over an aggregate ~70 kb of noncoding genomic space at the *LDLR* locus (S3 Table). As internal negative controls, we added 100 nontargeting gRNAs and 10 gRNAs targeting the AAVS1 'safe harbor' [23]. As internal positive controls, we added 10 experimentally validated [9] gRNAs apiece targeting the coding sequences of *LDLR* or its negative regulator *MYLIP*. We then synthesized a CRISPR library containing these gRNA sequences, verified library representation by deep sequencing, and generated and titered lentiviral stocks for pooled mutagenesis of target cell lines.

### Interrogation of putative *LDLR* CREs by CRISPR screening

To functionally interrogate putative *LDLR* CREs, we tested our CRISPR library in HuH7 cells (Fig 1A). This cell line has been established as a model of hepatocyte LDL catabolism [24] and, in our prior screen of the coding genome, exhibited clear dependence of LDL uptake on canonical regulators of *LDLR* expression including *SCAP*, *MBTPS1*, *MBTPS2*, and *MYLIP* [9]. For each of 4 independent biologic replicates, we transduced ~25 million HuH7 cells at a multiplicity of infection of approximately 0.5, such that most individual cells would not receive multiple gRNAs. After 2 weeks of passaging to allow for target site editing, we incubated pools of transduced cells with exogenous fluorescently conjugated LDL, detached cells, and performed flow cytometry to collect 10% subpopulations of edited cells with the greatest and least amount of LDL uptake. We then quantified the abundance of every gRNA in each sample by massively parallel sequencing and calculated gRNA enrichment between selected populations (S4 Table).

We assessed the technical performance of our screening pipeline by the segregation of control gRNAs. As expected, gRNAs targeting the coding sequence of *LDLR* and *MYLIP* were consistently depleted or enriched, respectively, in LDL$^{high}$ cells relative to LDL$^{low}$ cells, whereas gRNAs lacking a genomic target site or targeting the *AAVS1* locus were similarly represented in either population (Fig 1B). The segregation of these control gRNAs was reproducible in pairwise comparisons of independent biologic replicates (Pearson coefficients 0.86-0.96, S2 Fig).

Of the remaining gRNAs targeting putative *LDLR* CREs, the majority showed no significant difference between LDL$^{low}$ and LDL$^{high}$ populations (Fig 1C and S4 Table). Relatively few individual gRNAs were observed with significant enrichment in LDL$^{high}$ populations, suggesting a lack of repressive elements among the regions targeted by our library. In contrast, we

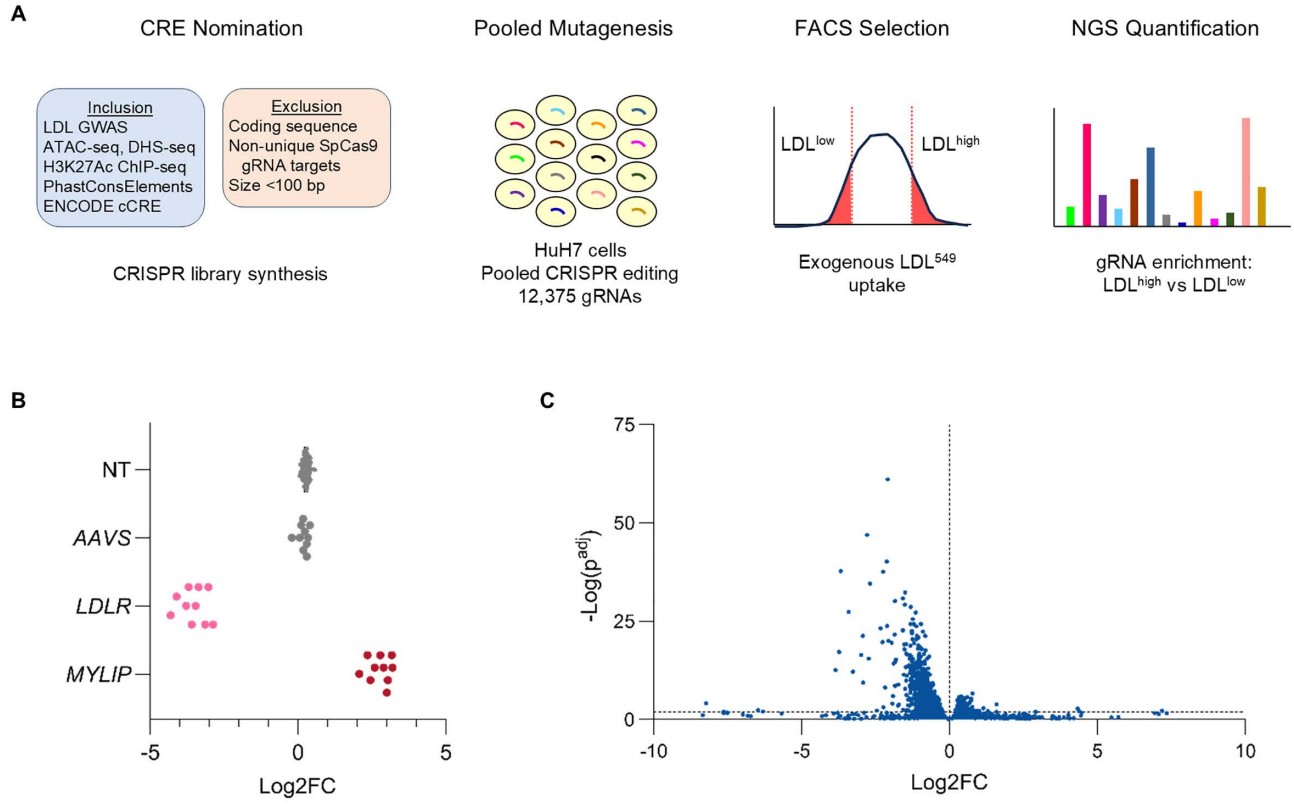

**Fig 1. CRISPR screen for noncoding regulators of *LDLR* expression. (A)** Schematic overview of CRISPR screen design with candidate CRE selection criteria and pooled targeting by CRISPR indel formation followed by FACS isolation of individual cells with aberrant LDLR activity and quantification of enriched gRNAs by next-generation sequencing (NGS). **(B)** Segregation between LDL$^{high}$ and LDL$^{low}$ populations of internal control gRNAs either lacking a genomic target sequence (NT) or targeting the *AAVS* safe harbor (both in gray), or targeting the coding sequences of *LDLR* (pink) or its negative regulator *MYLIP* (dark red). **(C)** Volcano plot of individual gRNAs targeting *LDLR* candidate *cis*-regulatory elements, with x-axis representing log2 fold-change in LDL$^{high}$ cells relative to LDL$^{low}$ cells and y-axis representing the negative log-transformed adjusted p-value as determined by DESeq2. Dashed lines indicate log2 fold-change of 0 and an adjusted p-value of 0.01. Control gRNAs displayed in (B) are omitted from **(C)**. Source data for CRE nomination and CRISPR screen results are provided in S3 and S4 Tables.

found 9.0% (1099/ 12,198) of gRNAs to exhibit significant depletion (adjusted p-value < 0.01) in LDL$^{high}$ cells, consistent with indel formation at their target sites leading to a decrease in *LDLR* expression. As expected, the strongest effects were observed for gRNAs predicted to generate DNA double-strand breaks (DSBs) near intron-exon junctions (IEJs), likely due to a disruption of RNA splicing and/or protein coding sequence by the resulting indels (S3 Fig). After filtering gRNAs to remove those predicted to generate DSBs within 100 bp of an intron-exon junction, 867 individual gRNAs remained that conferred a significant reduction in *LDLR* expression.

To aggregate and smooth individual gRNA enrichment scores into CRE peaks, we next analyzed sliding 100 bp windows centered on each gRNA in the library. Within each window, we identified all gRNAs predicted to generate a DSB and calculated the median p-value for the group (Fig 2 and S4 Table). Because the precision of this measurement depends upon the density of gRNAs in each window (S4A Fig), we established dynamic thresholds for significance based on the cutoff required to yield a false positive rate <10$^{-4}$ (i.e., < 1 expected false discovery among ~10,000 windows) with different numbers of random samplings of negative control gRNAs (S4B Fig). Windows meeting these thresholds that were contiguous were then merged into single CRE peaks. Application of this approach resulted in 25 significant and distinct CRE peaks spanning the *LDLR* locus (Fig 2 and S5 Table). Functionally important regions were skewed toward proximity to the

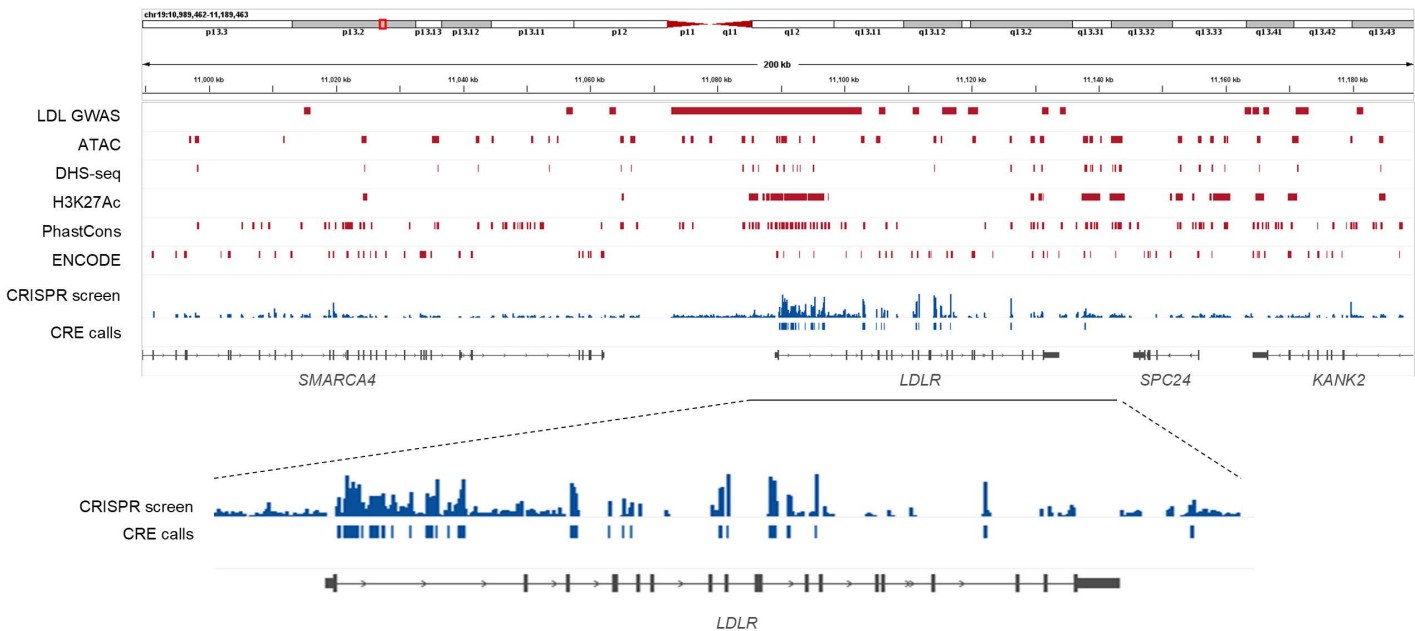

**Fig 2. Spatial organization of CRISPR screen results.** Alignment of genomic tracks for candidate CRE regions (red) grouped by the basis of their selection together with the CRISPR screen results (blue). Plotted values for the CRISPR screen results reflect the median negative log p-value over sliding 100 bp windows centered on each coordinate (top) and called CRE peaks (bottom) based on significance testing as described in Methods. Source data for CRISPR screen results are provided in S4 and S5 Tables.

*LDLR* TSS and, consistent with similar studies of other genes [25–27], demonstrated only partial overlap with the individual features used for CRE prediction.

### *LDLR* CREs colocalize with a subset of common variants highly associated with LDL cholesterol

We next visualized the enrichment of all individual gRNAs targeting the ~ 30 kb region containing all 72 variants highly associated with LDL by GWAS (Fig 3). Surprisingly, the entire region upstream of the *LDLR* promoter containing 56 of these variants showed minimal activity in our screen, with no significant CRE peaks identified by our analysis. We did not directly interrogate the *LDLR* promoter, as its overlap with an annotated antisense RNA led to its filtering from our candidate list. By contrast, many gRNAs targeting the first *LDLR* intron were significantly enriched in LDL$^{low}$ cells, suggesting that indel formation at their target sites caused a reduction in *LDLR* expression. While some of the active gRNAs targeted sequences near the first intron-exon junction, most targeted regions >500 bp from the first exon. The greatest activity was observed over two large and adjacent CRE peaks in the early first intron, one of which (hg38 Chr19: 11,090,422– 11,091,028) overlapped a single variant (rs59281581) highly associated with LDL.

We cross-referenced this functional map of the *LDLR* early first intron with other data sets that might further suggest *cis*-regulatory activity (Fig 3). Indeed, we found this cluster of activity in our screen to colocalize with open chromatin as indicated by ATAC-seq of human liver tissue, a distinctive enhancer-associated "peak-valley-peak" distribution for histone H3K27-acetylation [28] in human liver tissue, and an enrichment in JASPAR-predicted transcription factor binding motifs [29]. Included among the transcription factor binding motifs were two sites for SREBP binding, though most gRNAs that affected LDL uptake in this region targeted sites >100 bp from this motif. This region was also highly conserved among vertebrates, with PhastConsP scores approximating those observed for the nearby coding sequence of exon 1. Collectively, these findings suggest the presence of a conserved enhancer in the first *LDLR* intron that operates independently of direct SREBP binding.

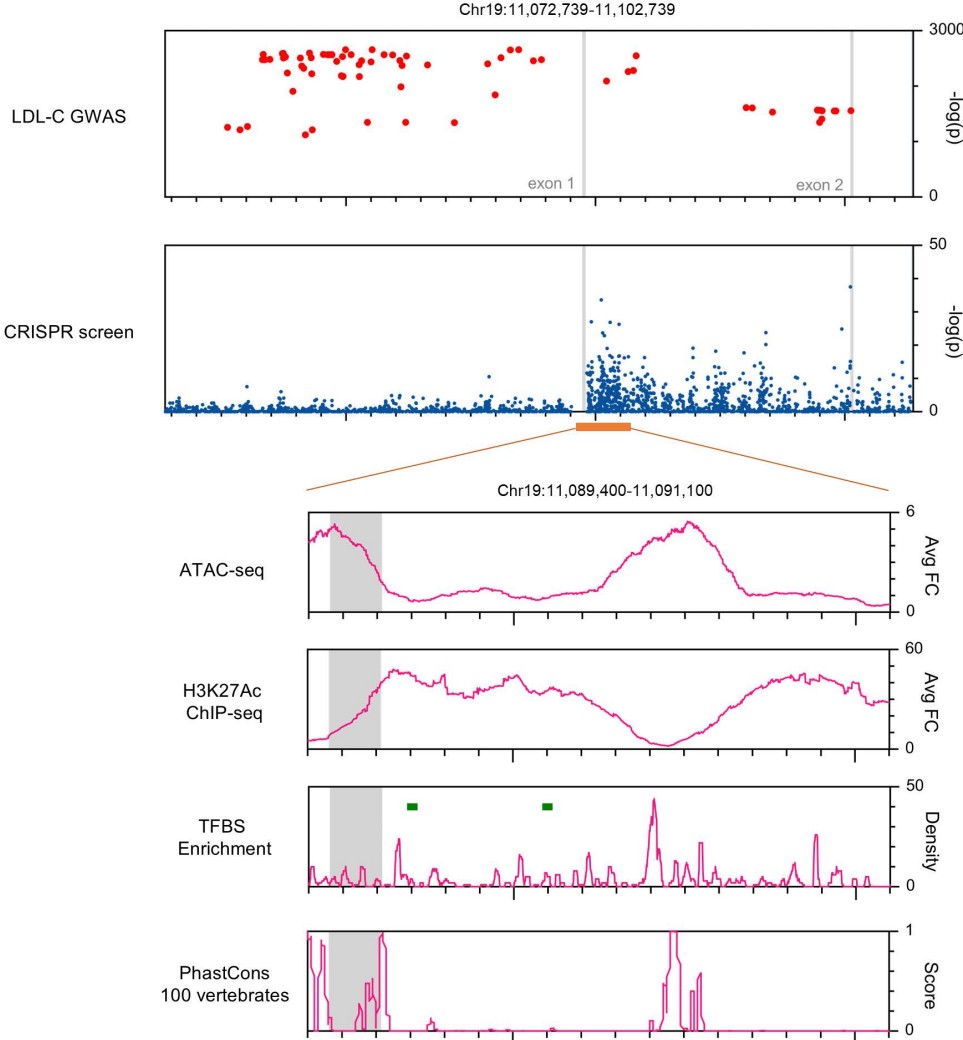

**Fig 3. Analysis of *LDLR* early first intron.** Candidate 30 kb interval spanning all common variants associated with LDL cholesterol with $p < 10^{-1000}$ by GWAS (red) aligned with CRISPR screen results (blue) for the negative log p-value for depletion in LDL^high relative to LDL^low cells of each individual gRNA in the screen arranged by the coordinate of its predicted double-strand break site. Associated tracks for this same region are displayed for average values from human liver tissue ATAC-seq and H3K27Ac ChIP-seq, the density of transcription factor binding motifs identified by JASPAR, and the evolutionary conservation scores across 100 vertebrates. Exons are shaded in gray. Green boxes on the TFBS plot correspond to canonical SREBP binding motifs. Individual gRNA screen source data are provided in S4 Table.

## Confirmation of enhancer activity in the first intron of *LDLR*

To validate the functional significance of the first intronic *LDLR* region to endogenous *LDLR* expression, we first selected 2 individual gRNAs that targeted this region and exhibited strong depletion in LDL^high cells in our screen. We cloned these sequences into individual lentiCRISPRv2 constructs, transduced and selected HuH7 cells, and performed phenotypic characterization of the resulting polyclonal populations. In comparison to control cells transduced with a nontargeting gRNA, cells with targeting of the *LDLR* first intronic region exhibited significant reductions in LDL uptake (Fig 4A) and *LDLR* mRNA levels (Fig 4B). Transduction of these constructs into HepG2 cells likewise resulted in a reduction in LDL uptake. This decrease was more apparent after culturing cells in lipoprotein-depleted conditions to stimulate increased

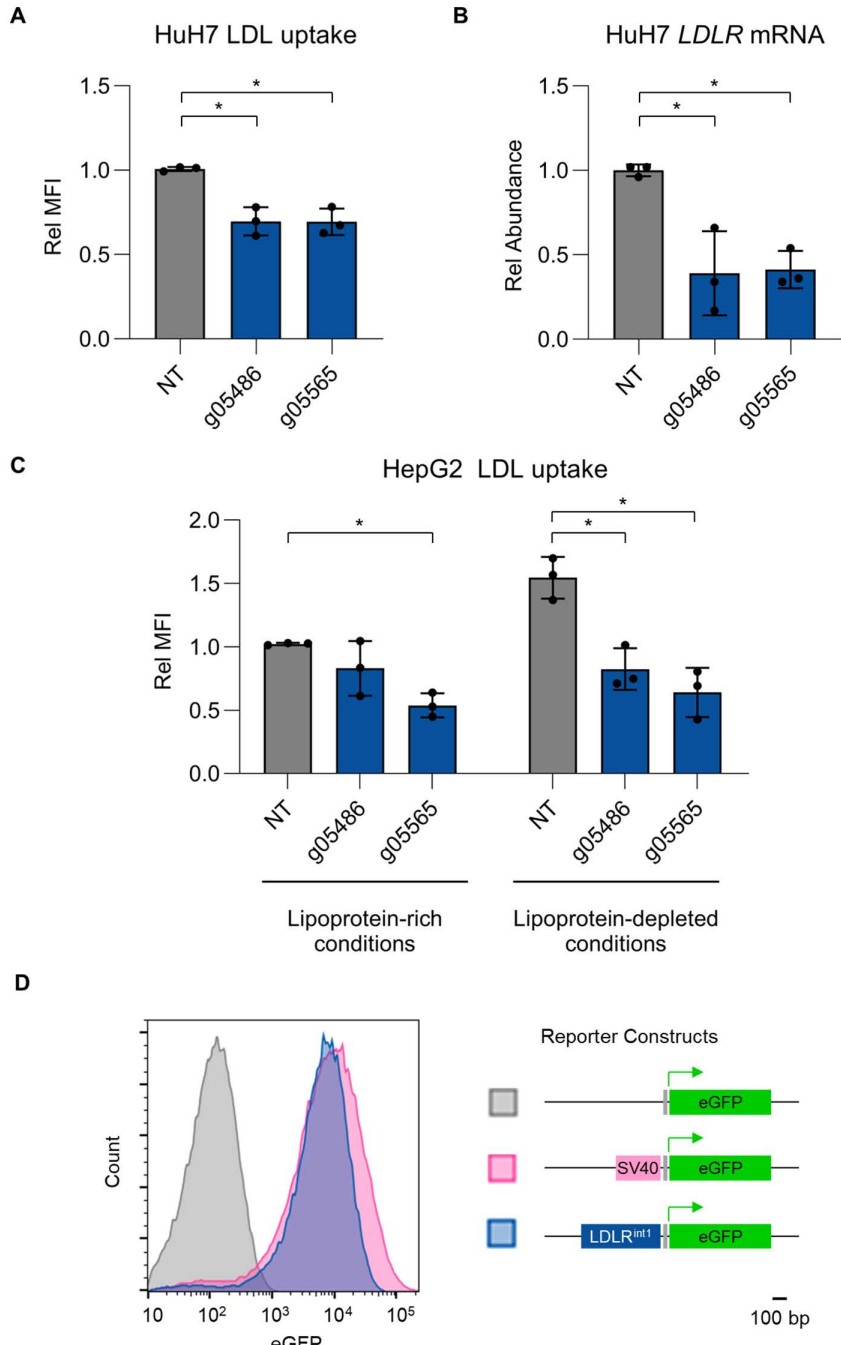

**Fig 4. Validation of enhancer activity in the first *LDLR* intron. (A)** Relative mean fluorescence intensity for LDL uptake by HuH7 cells transduced with a control nontargeting gRNA (gray) or individual gRNAs targeting the first *LDLR* intron that showed highly significant enrichment in the CRISPR screen (blue). **(B)** Relative *LDLR* mRNA levels as determined by qRT-PCR of mRNA preparations from the same HuH7 cells depicted in **(A)**. **(C)** Relative mean fluorescence intensity for LDL uptake by HepG2 cells transduced with the same individual gRNAs as in (A-B) and cultured in the presence of normal serum or lipoprotein-depleted serum for 48 hr prior to the LDL uptake assay. **(D)** Schematic representation of lentiviral reporter constructs containing an eGFP coding sequence with a minimal promoter and an upstream test sequence. Histograms depict eGFP fluorescence for cells transduced with each construct in parallel at equal multiplicity of infection. For all panels, individual data points represent independent biologic replicates and asterisks indicate $p < 0.05$ for the indicated sample relative to control cells transduced with a nontargeting gRNA by one-way ANOVA with Dunnett's correction for multiple hypothesis testing.

*LDLR* expression (Fig 4C), consistent with a lower baseline expression level of *LDLR* in HepG2 cells relative to HuH7 cells and with our prior observation that HepG2 cells exhibited a clearer dependence on established *LDLR trans*-regulators when cultured in lipoprotein-depleted conditions [9].

To further test for conventional enhancer activity in this region, we next PCR-amplified a portion of this region (hg38 Chr19:11,090,150–11,090,715) from HuH7 genomic DNA and inserted it upstream of a minimal promoter and eGFP coding sequence in a lentiviral reporter plasmid [30]. Analysis by flow cytometry revealed the *LDLR* first intronic sequence to confer a ~50-fold increase in eGFP fluorescence relative to the parental construct (Fig 4D). The magnitude of effect for the *LDLR* first intronic sequence was comparable to that observed for the positive control, a SV40 sequence that is well-characterized to exhibit strong enhancer activity [31,32]. Together with the individual gRNA validation described above, these findings confirm the presence of enhancer activity within the first intronic region identified by our CRISPR screen.

## Fine mapping of enhancer activity in the first intron of *LDLR*

Although our CRISPR screen revealed enhancer activity within the first intron of *LDLR*, interpretation of its precise localization was complicated by intrinsic limitations of the approach, including the irregularity of SpCas9 gRNA targets across the region, the expected variability in activity for different gRNAs within the pool, and the expected heterogeneity of indel sizes arising at target sites. Therefore, to fine map enhancer activity in this region we instead developed a massively parallel reporter assay (MPRA). In this approach, large pools of candidate *cis*-regulatory sequences are tested in parallel for their ability to augment expression of a reporter gene [33]. We designed an oligonucleotide pool with 74 overlapping tiles that spanned an 860 bp interval with the greatest activity in our CRISPR screen (Fig 5A and S5 Table). These included tiles of variable sizes (172, 129, 86, and 43 bp) in order to define the minimal region sufficient for enhancer activity. We designed our library to represent the major allele in gnomAD [34], notably including a common single nucleotide insertion (rs59281581) that has an estimated allele frequency of 99.8% but is not present in the hg38 reference sequence. We also included 3 positive and 3 negative internal control sequences that did or did not demonstrate *cis*-regulatory activity in a prior MPRA in HepG2 cells [35]. We amplified these sequences from oligonucleotide pools and cloned them into the pLS-SceI reporter backbone with incorporation of a random 15N barcode in the 5' UTR of each construct. Sequencing of the resulting plasmid pool confirmed the synthesis of 69/74 expected tiles with a median of 2653 independent 5'UTR barcodes for each (S5A Fig and S7 Table). Expected tiles that were absent all shared the same 86 bp sequence with 78% GC content and a predicted hairpin structure that likely precluded its efficient synthesis, amplification, and insertion into the reporter plasmid (S5B Fig).

To functionally interrogate each tile, we then transduced HuH7 cells with the MPRA library, harvested genomic DNA and mRNA 2 days later, and performed massively parallel sequencing to quantify 5'UTR barcode abundance in each sample (S8 Table). Analysis of cDNA/gDNA ratios confirmed an increase in expression for positive controls relative to negative controls (Fig 5B and S9 Table). Of the other 69 test sequences, most exhibited no significant increase in activity relative to negative controls. A subset of test sequences however did show robust activity comparable to or exceeding that of the positive control sequences. The expression values for individual tiles in the MPRA library exhibited very high reproducibility between individual replicates (S6 Fig, $r > 0.98$, $p < 0.001$). To examine the generalizability of these findings across cell lines, we repeated the same MPRA approach in HepG2 cells, again finding discrimination of control sequences and a very high degree of concordance for the activity of test tiles between HuH7 and HepG2 cells (Fig 5C, $r > 0.99$, $p < 0.001$).

Strikingly, the individual tiles that exhibited the greatest enhancer activity in our HuH7 and HepG2 MPRA studies all overlapped the same region (Fig 5D). This region was approximately centered on the ATAC-seq peak and the valley of the H3K27 acetylation ChIP-seq peak-valley-peak in this region, was highly conserved among vertebrates, and contained several motifs for transcription factor binding as well as one of the common genetic variants (rs57217136) strongly associated with LDL cholesterol by GWAS [12–14] (S7 Fig). Of the implicated transcription factors, most exhibited detectable expression in HuH7 and HepG2 cells by publicly available RNA-seq data (S8 Fig).

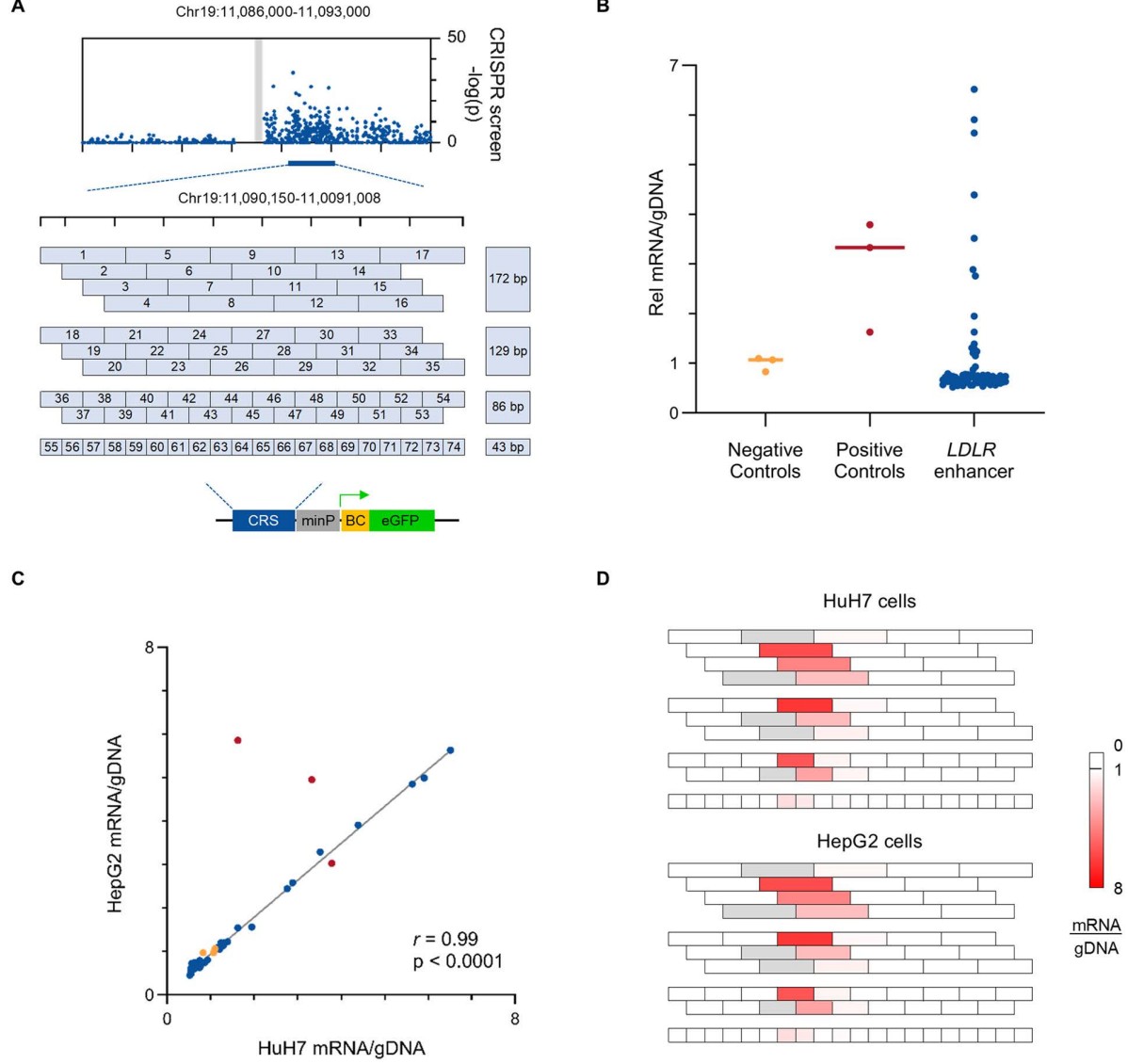

**Fig 5. Fine mapping enhancer activity with a massively parallel reporter assay. (A)** Design of the MPRA library containing overlapping tiles of noncoding sequence spanning different lengths of the candidate enhancer interval. **(B)** Relative reporter expression in HuH7 cells associated with each tile, derived from mapped sequencing reads of cDNA relative to genomic DNA. **(C)** Correlation of reporter expression conferred by each sequence in the MPRA library in HuH7 cells and HepG2 cells. Red and orange data points represent internal positive and negative controls, respectively, and blue data points represent candidate *LDLR* enhancer tiles. **(D)** Spatial organization of enhancer activity in each MPRA. A heat map is displayed for each tile visualized in **(A)**, with greater enhancer activity represented by a deeper shade of red. Tiles with inadequate sequencing coverage in the plasmid pool are shaded in gray. Source data for MPRA analysis are provided in S9 Table.

## Validation of fine-mapped enhancer activity

To validate the presence of enhancer activity in the narrower region identified by our MPRA studies, we next generated reporter constructs harboring 3 different fragments of the putative fine-mapped enhancer in both the forward and reverse orientations (Fig 6A). For 2 of the 3 fragments (containing tiles 6 and 42), we found enhancer activity to fully recapitulate that of the original full length construct; in each case, this activity was observed in either orientation, consistent with

PLOS Genetics

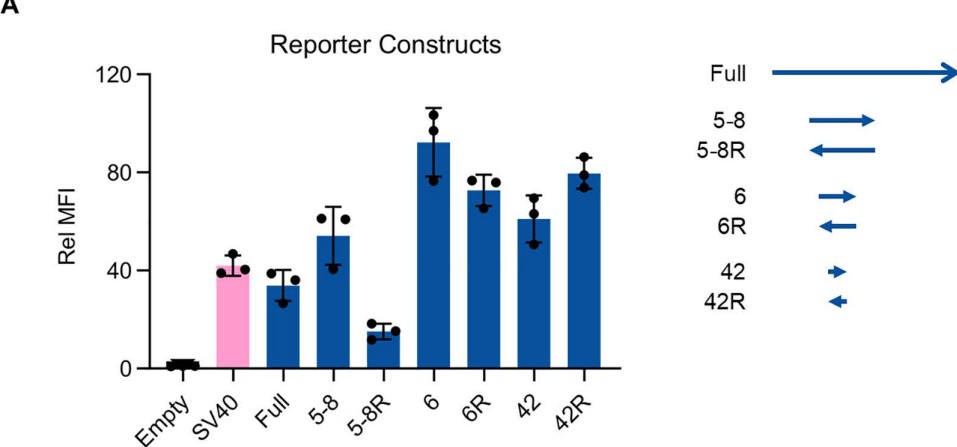

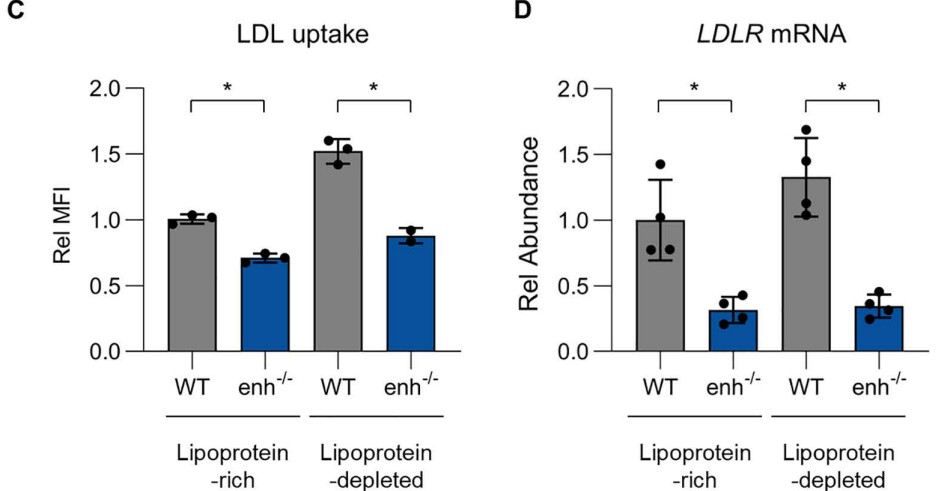

**Fig 6. Validation of fine-mapped enhancer activity. (A)** Mean fluorescence intensity (MFI) of eGFP relative to empty vector control for individual reporter constructs containing either the full length region containing the *LDLR* enhancer or the individual fragments identified by the MPRA in the forward or reverse orientations. **(B)** Genotype of a single cell clone recovered after transfection with a pair of gRNAs predicted to generate double-strand breaks between the nucleotides highlighted in red. **(C-D)** Uptake of fluorescent LDL (C) and *LDLR* mRNA levels (D) for the enhancer-deleted clonal population relative to parental wild-type cells, following culture of each cell line in lipoprotein-rich or lipoprotein-depleted media. Individual data points represent independent biologic replicates and asterisks indicate $p < 0.05$ by Student's t-test.

classical enhancer activity. For the 3rd and largest fragment containing tiles 5–8, enhancer activity was present in either orientation though partially reduced in the reverse orientation, potentially related to an increased distance between the core active sequence and the minimal promoter.

To test the functional significance of this fine mapped enhancer activity in the context of the endogenous *LDLR* locus, we next transduced HuH7 cells with constructs expressing SpCas9 and a pair of gRNAs whose target sites flanked the minimal tile containing complete enhancer activity by MPRA testing. Genotyping of resulting single cell clones identified a cell line with compound heterozygous loss of the fine-mapped enhancer in both alleles (Fig 6B). Phenotypic characterization of this cell line confirmed a defect in LDL uptake (Fig 6C) and a reduction in *LDLR* mRNA levels (Fig 6D) under both lipoprotein-rich and lipoprotein-depleted conditions. These findings are consistent with the results of the reporter assay testing and confirm the presence of an enhancer in the first intron of *LDLR*.

## Discussion

Historically, the criteria for identifying *cis*-regulatory elements have evolved along with the tools available for their interrogation [7]. Early definitions of enhancers referred to the ability of DNA sequences to increase the expression of reporter plasmids [31,36]. Over time, descriptions of enhancers began to incorporate biochemical features such as hypersensitivity to DNAse treatment (reflecting a lack of nucleosomes), an association with distinctive histone marks, and the presence of transcription factor binding sites. Although each of these features are useful, they remain imprecise tools for CRE identification. Numerous counterexamples exist both for *bona fide* enhancers that lack classical predictive features, and for DNA regions that were suspected to be enhancers based on these criteria but exhibited no such activity upon functional testing [7,37–40]. Furthermore, biochemical predictions of enhancer activity often depend on coarse signals (e.g., broad peaks) that preclude precise mapping to specific boundaries.

Recently, advances in genome engineering have revolutionized the field of functional genomics. While high-throughput CRISPR screening has been mainly applied to the coding genome, studies that have extended this approach to the noncoding genome have consistently demonstrated its power to define the functional significance of CREs in their native genomic context [25–27,41–43]. Consistent with these studies, we now report our identification of 25 functionally significant *LDLR* CREs by high-throughput CRISPR screening in HuH7 cells. The fidelity of our results is supported by our clear discrimination of control gRNAs, the reproducibility in gRNA enrichment across independent biologic replicates, the spatial clustering of our screen hits, and by the empiric validation of a region in the first intron in follow up studies.

We found that many noncoding regions of the *LDLR* locus with predicted CRE activity exhibited no impact on *LDLR* expression when targeted by CRISPR-mediated indel formation. This included the entire 5' intergenic region upstream of *LDLR* that was tested in our screen. We did not directly target the *LDLR* promoter, wherein certain sequences with functional significance have been identified by single variant [44–47] and MPRA [48] studies. Other noncoding CRISPR nuclease screens have identified functionally significant CREs in the 5' intergenic region of their tested genes [25–27,43], suggesting that our findings do not represent an inherent limitation of the approach but rather a gene-specific phenomenon. It has been reported that enhancers for genes with tissue-specific activity are highly enriched in intronic regions, whereas enhancers for housekeeping genes are more commonly intergenic [49]. Therefore, while different regions in the 5' intergenic region of *LDLR* exhibit features predictive of CRE activity, they may be either nonfunctional or have a functional impact on other genes like the nearby *SMARCA4*.

Intriguingly, the majority of the ~30 kb interval spanning 72 noncoding variants with the strongest LDL cholesterol associations at the *LDLR* locus by GWAS [12–14] overlapped the 5' intergenic region lacking *LDLR* regulatory activity in our screen. This observation suggests that many of these variants may not directly influence *LDLR* expression but rather correlate with LDL due to their linkage disequilibrium with 1 or more other causal variants. However, there are several caveats to this interpretation that warrant consideration. First, enhancer activity may be highly context-dependent and not recapitulated by the conditions of our screen; that is, a noncoding region may show no effect in our study yet still play an important role in *LDLR* expression in other cell types and/or in response to different developmental or environmental cues. Second, pooled CRISPR screening is inherently limited in detecting sequences that contribute to gene expression in aggregate but do not cause a discernible phenotype when disrupted individually, as may be the case for CREs containing

redundant transcription factor binding sites. Third, our ability to disrupt candidate CREs is limited by the presence of SpCas9 PAM motifs (with an average density of 1 per 8 bp across the human genome) in regions of interest. Similarly, individual gRNAs may vary substantially in their relative activities. A given CRE therefore may be missed by our approach if it exists in a region with a paucity of gRNA targets or with target sequences for gRNAs with low activity. Finally, the edits that arise at CRISPR target sites are heterogeneous among individual cells transduced with the same gRNA. The most common indels resulting from CRISPR are typically <50 bp, which may not sufficiently disrupt larger CREs. On the other hand, larger indels can also occur [50–53]. We cannot exclude the possibility that some CREs identified in our study may represent false positives owing to gRNA enrichment being driven by the subset of cells harboring these larger indels, especially for target sites in relative proximity to an intron-exon junction.

Alternative approaches for functional screening of candidate CREs include CRISPR inhibition or activation, in which catalytically dead Cas9 is fused to effector domains that modulate transcription, and ScanDel, in which pairs of gRNAs mediate longer deletions [7,54,55]. These strategies may overcome some of the limitations of CRISPR nuclease screening described above, as they may result in more consistent perturbations for each gRNA and broader disruptions that affect multiple transcription factor binding sites.

The limitations of CRISPR nuclease screening notwithstanding, our study expands the functional understanding of the noncoding landscape at the *LDLR* locus. In addition to identifying 25 functionally significant CREs across the *LDLR* locus, for 1 in the first intron we validated and fine mapped enhancer activity to a 129 bp region that was highly conserved, associated with biochemical hallmarks of *cis*-regulatory activity, and enriched in transcription factor binding motifs. This region also contains a single common genetic variant (rs57217136) that is strongly associated with human LDL-cholesterol and in linkage disequilibrium with other top LDL-associated variants at the *LDLR* locus. Intriguingly, a prior study found the minor allele of rs57217136 to increase reporter gene expression in HuH7 cells, but in contrast to our results did not detect enhancer activity for the major allele [15]. The basis for this difference is unclear but may be related to details of experimental design including the specific reporter constructs and readouts used in each study.

The first intronic enhancer identified in our study does not contain a canonical sterol response element, the binding site for SREBP transcription factors and the primary established mechanism for *LDLR* gene regulation. Instead, this region contains putative binding sites for several other transcription factors including: Krüppel-like factors (KLFs), some of which have been previously implicated in atherogenesis and hepatic metabolism [56,57]; specificity proteins (SP), which are evolutionarily related to KLFs and one of which (SP1) has been shown to act synergistically with SREBP at the *LDLR* promoter [58,59]; and glioma-associated oncogenes (GLI), which are involved in hedgehog signaling [60] but have not been directly linked to lipoprotein metabolism or atherosclerosis. Although none of these transcription factors were identified in our prior LDL uptake screen of the coding genome [9], functional redundancy among members of the same family could mask the development of a phenotype for any single gene disruption. A previous study also reported binding by RAR and STAT1 to this region for the minor allele of rs57217136, though no binding or enhancer activity was detected for the reference allele [15]. It is also possible that other proteins, including SREBP, may bind to flanking regions at the endogenous locus and contribute to enhancer activity. Although we did not detect activity in the immediate surrounding regions by MPRA, this does not exclude a potential synergistic influence of these flanking regions. Further work will be necessary to dissect the relative contributions of these and other DNA-binding proteins to enhancer activity in the first *LDLR* intron.

## Materials and methods

### Cell lines and reagents

HuH7 and HepG2 cells (ATCC, Manassas VA) were cultured in DMEM supplemented with 10% fetal bovine serum, 10 U/mL penicillin, and 10 µg/mL streptomycin (Thermo Fisher Scientific, Waltham MA) in a humidified 5% $CO_2$ chamber at 37°C. Cell lines were periodically tested for mycoplasma contamination and verified by microsatellite genotyping. Oligonucleotide sequences (Integrated DNA Technologies, Coralville IA) are provided in S10 Table. CRISPR constructs

were generated by ligation of phosphorylated and annealed oligonucleotides into BsmBI-digested pLentiCRISPRv2 [61] (Addgene, Watertown MA, #52961) or pX459 [62] (Addgene #62988) as previously described [63]. Reporter constructs were generated by PCR amplification of the indicated noncoding sequence from genomic DNA or a plasmid template with primers designed to provide flanking homology arms for HiFi assembly into AgeI/SbfI-digested (New England Biolabs, Ipswich MA) pLS-SceI$^{30}$ (Addgene #137725) as previously described [30]. A plasmid with the same backbone and a SV40 enhancer sequence was also obtained as a positive control (pLS-SV40-mP-EGFP, Addgene #137724). Plasmids were verified by Sanger sequencing or whole plasmid sequencing and lentiviral stocks were prepared in coordination with the University of Michigan Vector Core.

## Candidate CRE nomination and analysis

Summary statistics were downloaded from the Global Lipid Genetics Consortium multi-ancestry GWAS of LDL cholesterol levels in 1,654,960 individuals [14]. Individual SNP associations were visualized together with their linkage disequilibrium with the lead SNP using LocusZoom [64]. The 30 kb interval spanning all SNPs with $p < 10^{-1000}$ was selected for saturation editing. To identify more distal variants with independent associations with LDL levels, we performed a conditional analysis on the lead SNP (rs73015024) for all variants within 500 kb of the *LDLR* TSS with an allele frequency >1% in individuals of European ancestry using GCTA-COJO [65]; from this analysis we targeted a 1 kb interval centered on each of 29 SNPs independently associated with LDL. Additional candidate *LDLR* CREs were identified by analysis of genomic tracks downloaded from the ENCODE Project [22] and UCSC Table Browser [66] and are graphically depicted in Fig 2. Identifying information for individual tracks is provided in S11 Table and the basis for selection of individual gRNAs is provided in S3 Table. In brief, regions of open chromatin in hepatocytes were identified by colocalization with peaks for ATAC-seq or DHS-seq from human liver tissue and/or HepG2 cells [17,18]. Regions associated with histone marks of enhancer activity were identified by colocalization with peaks for H3K27Ac ChIP-seq from human liver tissue and/or HepG2 cells [19]. Evolutionarily conserved regions were identified by downloading the phastConsElements30way table from the 30 Primates track of UCSC Genome Browser, based on a previously described phylogenetic reconstruction [67]. Regions were excluded if they colocalized with exons in the NCBI RefSeq track of UCSC Table browser or if they spanned <100 bp of genomic space. All SpCas9 target sites within 500 MB of the *LDLR* TSS were downloaded from the CRISPR Targets track of USCS Table Browser and filtered for those whose predicted double strand break location overlapped the candidate CRE blocks identified above. For subsequent analyses of CRISPR screen results, additional data sets were downloaded from the ENCODE Project and USCS Table Browser. Predicted transcription factor binding sites were obtained from UCSC Table Browser track JASPAR TFBS, filtered for quality scores >=400, and enumerated at each chromosomal coordinate in the candidate enhancer interval. The expression level of different transcription factors in HuH7 and HepG2 cells was derived from RNA-seq data in DepMap release 24Q4 [68]. Evolutionary conservation was analyzed by downloading the PhastCons tables from the tracks for 30 Primates and 100 Vertebrates in UCSC Table Browser; rolling averages across a 11 bp window centered on each chromosomal coordinate were calculated for each table and are plotted in Fig 3.

## CRISPR library synthesis

A custom CRISPR library was synthesized as previously described [9,69] to encode 12,375 unique gRNAs targeting the candidate CRE blocks identified above as well as 100 nontargeting negative control sequences, 10 negative control sequences targeting the *AAVS1* genomic "safe harbor", and 10 positive control sequences apiece targeting the coding sequences of *LDLR* and *MYLIP* based on prior functional data [9]. Briefly, a pooled library of oligonucleotides was obtained that contained each gRNA sequence flanked by invariant annealing sites for PCR amplification (CustomArray, Bothell WA). This library was amplified using primers that provided additional flanks for HiFi assembly into BsmBI-digested pLentiCRISPRv2. Assembly products were electroporated into Endura electrocompetent cells (Lucigen, Middleton WI) and plated on 245 mm bioassay plates of LB-agar. Serial dilutions of electroporated cells were also prepared and colonies

enumerated to confirm > 50X library coverage and a > 20:1 ratio for assembly reactions relative to control reactions with water in place of the oligonucleotide pool for initial PCR amplifications. Plasmid DNA was harvested from the resulting colonies and library representation was verified by massively parallel sequencing with an Illumina MiSeq single end 1x150 read and mapping individual reads to expected gRNA sequences using PoolQ as previously described [70].

## CRISPR screen for modifiers of LDL uptake

For each of 4 independent biologic replicates, approximately ~50 million HuH7 cells were distributed across 8 separate 15 cm diameter plates and transduced with the CRISPR lentiviral library at an estimated MOI of 0.5. Transduced cells were selected with 3 µg/mL puromycin beginning on day 1 post-transduction and continuing until control non-transduced cells were no longer viable. Selected cells were passaged every 2–3 days as needed, maintaining > 1000X library coverage of total cell numbers with each passage. On day 14, cells were incubated with 4 µg/mL DyLight[549]-conjugated LDL (Cayman Chemical, Ann Arbor MI) in serum-free DMEM for 1 hr, harvested, and sorted by flow cytometry into subpopulations with the top and bottom 10% of LDL uptake. Genomic DNA was purified from each subpopulation using a DNEasy kit (Qiagen, Hilden Germany) and the integrated gRNA sequences were PCR amplified with incorporation of sample barcodes into the primers. Analysis of gRNA abundance between samples was performed by sequencing amplicons with an Illumina NextSeq paired end 2x100 read, mapping and demultiplexing reads with PoolQ, and assessing gRNA enrichment with DESeq2 [71]. After determining the confounding influence of proximity to an *LDLR* IEJ on individual gRNA effects, post hoc filtering was performed to remove those gRNAs predicted to generate a DSB within 100 bp of an IEJ. To aggregate the effects of gRNAs targeting the same region, 100 bp rolling windows were created that were centered on each individual gRNA target site; every gRNA predicted to generate a DSB within each window was identified and the score of the window was determined by the median p-value for all gRNAs for the group. To identify dynamic thresholds of significance based on the number of gRNAs within a given window, 50,000 simulations apiece were performed with 1–50 random samplings of the 100 nontargeting negative control gRNAs analyzed in the library. The p-value threshold that resulted in less than 0.01% false positives (i.e., sets of nontargeting gRNAs that would be considered significant) was then calculated. A given window was considered significant if the median p-value for gRNAs in that window was less than the p-value cutoff for the same number of samplings of the nontargeting gRNA set. Activity scores are provided together with the density of gRNAs and the associated significance threshold for each window in S4 Table. Overlapping windows meeting the threshold for significance were merged into single CRE peaks, which are listed along with their coordinates in S5 Table.

## Analysis of *LDLR* expression and LDL uptake

For comparison between control cells and cells with the indicated genetic perturbations, LDL uptake was assayed by incubating monolayers of cells in 6 well plates with 4 µg/mL of DyLight[549]-conjugated LDL in serum-free DMEM for 1–2 hrs at 37°C, washing with PBS, harvesting cells, and analyzing mean fluorescence intensity by flow cytometry. To adjust for the confounding influence of cell density on LDL uptake per cell, a standard curve was prepared by plating control cells at a range of densities and estimating cell density by the number of flow cytometry events for a fixed volume of cell suspension. LDL uptake was then normalized relative to the expected fluorescent intensity at a given cell density. For measurements of *LDLR* expression, cellular RNA was harvested with an RNEasy Plus Mini kit (Qiagen), converted to cDNA using Superscript III Reverse Transcriptase (Thermo Fisher), and analyzed by qPCR with Power SYBR Green (Thermo Fisher), gene-specific primers (S9 Table), and QuantStudio 5 (Thermo Fisher).

## Enhancer reporter assays

For initial testing of enhancer activity in the first *LDLR* intron, PCR was performed with PrimeSTAR GXL polymerase (Takara Bio, Kusatsu Japan) and the primers listed in S10 Table to amplify the target region from HuH7 genomic DNA.

Additional rounds of PCR were then used to append vector homology arms and incorporate a random 15N barcode into the 5' UTR of the reporter construct. HiFi DNA assembly (New England Biolabs) was performed to insert the amplicon into AgeI/SbfI-digested pLS-SceI. Subsequent constructs were generated by similarly amplifying the region of interest from the parental enhancer construct. Lentiviral titers were calculated by transducing HuH7 cells at 20% confluence in 6 well plates, harvesting genomic DNA 2 days later, and performing qPCR to quantify the integrated lentiviral DNA relative to plasmid and host genomic DNA and translate these values to an estimated multiplicity of infection as previously described [30]. Transductions were then repeated with adjustment of each lentiviral stock volume to achieve the same MOI. Cells were harvested at 2 days post-transduction and analyzed for eGFP mean fluorescence intensity by flow cytometry.

**MPRA library synthesis**

Individual overlapping tiles of varying sizes in the candidate region of *LDLR* intron 1 were selected for fine mapping as indicated in S6 Table. The overall strategy for library synthesis was performed as previously described [30], with adaptations as indicated below. Briefly, an oligonucleotide pool was obtained (Integrated DNA Technologies) that included each candidate tile flanked by invariant sequences that served as annealing sites for amplification. Two rounds of PCR were performed with PrimeSTAR GXL polymerase using an annealing temperature of 60°C and the primers listed in S10 Table that added additional flanks for HiFi assembly and incorporation of a random 15N barcode into the 5' UTR of the reporter construct. The number of cycles was optimized by analysis of PCR amplicons on a Bioanalyzer DS1000 chip (Agilent Technologies, Santa Clara CA). Amplicons were purified with a QIAquick PCR Purification kit (Qiagen) and a total of 50 ng were combined with 200 ng of AgeI/SbfI-digested pLS-SceI for HiFi DNA assembly (New England Biolabs) at 50°C for 1 hr. The assembly products were then column purified and electroporated into Stbl4 *E. coli* cells (Thermo Fisher) using a MicroPulser (Bio-Rad Laboratories, Hercules CA) with setting Ec1. Electroporated cells were spread onto 245 mm LB-agar plates. Total colony counts were estimated by also plating serial dilutions of the electroporated cell suspension in parallel. Following verification of >100X colony counts relative to library size with a > 20:1 ratio of colonies on assembly plates relative to control reactions containing digested vector alone, resulting colonies were scraped and plasmid DNA extracted using a Plasmid Plus Midi kit (Qiagen). The representation of the candidate tiles in the resulting plasmid pool was then tested by massively parallel sequencing. Amplicon libraries were prepared by 2 rounds of PCR using the primers indicated in S10 Table to add flanking sample barcodes and sequencing adaptors. Optimization of PCR cycle number and assessment of amplicon library quality were again performed with a Bioanalyzer DS1000 chip. Sequencing was performed on an Illumina MiSeq instrument using the custom primers indicated in S10 Table. CRS sequences were extracted from reads 1 and 2 and associated with 15 bp CRS barcodes in index read 1. Individual barcodes were filtered to remove any with a read count <10 or with <99% association to the same unique CRS. The resulting array of barcode-CRS associations was used as a reference map for subsequent MPRA testing and is provided in S7 Table. For expected CRSs not detected in the plasmid library, a shared sequence was analyzed for predicted secondary structure by Unafold [72]. Lentiviral stocks were prepared from plasmid pools and titered by transducing HuH7 cells with a range of virus volumes and quantifying the resulting MOI by qPCR from genomic DNA with primers for integrated lentiviral sequence, plasmid backbone, and host genomic DNA as described above.

**Massively parallel reporter assay**

HuH7 and HepG2 cells were seeded in 6 well plates at an estimated 20% confluence, with 5 wells used for independent transductions with the MPRA library at an estimated MOI of 5 and the remaining well serving as an uninfected control. Two days later, cells were harvested and split in half for extraction of genomic DNA and RNA using DNeasy and RNeasy Plus Mini kits, respectively (Qiagen). For each sample, 10 µg of RNA was treated with 10 units of Turbo DNAse (Thermo Fisher) for 30 mins at 37°C followed by addition of DNAse Inactivation Reagent for 5 mins at room temperature and conversion of 1.6 µg RNA to cDNA using the gene-specific primer P7-pLSmP-ass16UMI-gfp and Superscript III First-Strand

Synthesis Kit (Thermo Fisher) per manufacturer's instructions. Genomic DNA and cDNA samples were then used as templates for PCR amplification of CRS barcode sequences with primers providing additional barcodes for sample identity and Illumina adapters. The abundance of each CRS barcode in each sample was assessed by custom sequencing using the primers listed in S10 Table on an Illumina NextSeq instrument. CRS barcode sequences were extracted from individual reads, filtered to remove any with <10 counts across all samples, demultiplexed by the sample barcode in index read 2, mapped to their associated CRSs, normalized to counts per million, and aggregated across CRS barcodes mapping to the same CRS. The relative expression conferred by each CRS was inferred by calculating the average ratio of normalized counts associated with each CRS in cDNA versus genomic DNA for each individual replicate. Normalized read counts for each CRS are provided in S8 Table and expression analysis is provided in S9 Table.

### Enhancer deletion

HuH7 cells were co-transfected with an equal mixture of pX459 constructs harboring individual gRNAs targeting sequences upstream and downstream of the fine-mapped *LDLR* enhancer (S10 Table) using Lipofectamine LTX (Thermo Fisher) per manufacturer's instructions. Puromycin selection was applied from days 1–3 post-transfection at which time no viable cells remained for control untransfected cells. Puromycin-resistant transfected cells were detached into suspension and diluted for single cell cloning into 96 well plates. Wells containing a single focus of cell proliferation were subsequently expanded and genotyped by extracting genomic DNA using a DNEasy kit (Qiagen), PCR amplifying the target region with PrimeSTAR GXL polymerase (Takara Bio) and primers flanking the predicted double-strand break site (S10 Table), performing amplicon sequencing (Plasmidsaurus, Eugene OR), and aligning the resulting reads using the Integrated Genomics Viewer [73]. A clone harboring compound heterozygous deletion of the enhancer was selected for subsequent analysis with LDL uptake and *LDLR* expression as described above.

## Supporting information

**S1 Fig. LDL cholesterol-associated variants at the *LDLR* locus.** (A) LocusZoom plot of *LDLR* variant associations with LDL cholesterol, with individual SNPs shaded by their linkage disequilibrium with the lead SNP. (B) Genomic tracks of regions targeted by the CRISPR library, including a block containing all variants identified in (A) and 1 kb intervals centered on other SNPs within 500 kb of the *LDLR* TSS and an independent association with LDL cholesterol after conditioning on the lead SNP. Source data is provided in S1 and S2 Tables.
(TIF)

**S2 Fig. Reproducibility of control gRNA segregation.** Pairwise correlations and Pearson coefficients for the log2 fold change of individual control gRNA abundance in LDL$^{high}$ vs LDL$^{low}$ cells across 4 independent replicates of the CRISPR screen. Dark red data points correspond to gRNAs expected to be enriched in LDL$^{high}$ cells due to their disruption of the *MYLIP* coding sequence. Pink data points correspond to gRNAs expected to be depleted in LDL$^{high}$ cells due to their disruption of the *LDLR* coding sequence. Gray data points correspond to gRNAs expected to have no effect on LDL uptake related to their targeting of the AAVS safe harbor region.
(TIF)

**S3 Fig. Relationship of gRNA enrichment with proximity to intron-exon junctions.** Violin plots for average log2 fold-change gRNA abundance in LDL$^{high}$ relative to LDL$^{low}$ cells for individual gRNAs grouped by their proximity to the nearest *LDLR* intron-exon junction. Solid and dashed lines represent the median and quartiles, respectively, for each group.
(TIF)

**S4 Fig. Dynamic significance thresholds for rolling 100 bp window analysis.** Median p-values were calculated for the indicated number of random samplings of negative control gRNAs from the library, modeling the range in gRNA density

over 100 bp rolling windows across the interrogated genomic regions. (A) The standard deviation of this calculation over 50,000 simulations is plotted as a function of the number of gRNA samplings. (B) The -log(p) threshold necessary for a false positive rate $<10^{-4}$ (i.e., the 99.99th percentile highest value from the simulations) is plotted as a function of the number of gRNA samplings.
(TIF)

**S5 Fig. Analysis of MPRA library synthesis.** (A) The number of individual 5' UTR barcodes detected in association with each expected *cis*-regulatory sequence in the MPRA library. Source data is provided in S7 Table. (B) Predicted hairpin structure in the sequence common to all expected *cis*-regulatory sequences absent from the plasmid library.
(TIF)

**S6 Fig. Reproducibility of MPRA results.** Pairwise correlations and Pearson coefficients for the cDNA/gDNA ratios of individual tiles in the MPRA library across 5 independent replicates in HuH7 cells. Red and orange data points represent internal positive and negative controls, respectively, and blue data points represent candidate *LDLR* enhancer tiles.
(TIF)

**S7 Fig. Features of the fine-mapped first intronic *LDLR* enhancer.** (A) Genomic tracks from Fig 2 overlaid with light blue shading to indicate the position of the MPRA fine-mapped enhancer interval. (B) Genomic sequence of the fine-mapped interval aligned with positions of human genetic variants of interest, a heat map of PhasConsP evolutionary conservation scores among 100 vertebrates and 30 primates (deeper shades of purple indicate greater conservation), and transcription factor binding motifs identified by JASPAR.
(TIF)

**S8 Fig. Expression analysis of predicted enhancer-binding proteins.** RNA-seq expression levels in HuH7 and HepG2 cells for each gene encoding a protein with a predicted binding motif in the fine-mapped *LDLR* enhancer. Values are derived from DepMap release 24Q4.
(TIF)

**S1 Table. LDL cholesterol GWAS associations.** Common genetic variants at the *LDLR* locus associated with LDL cholesterol at a significance threshold of $p < 10^{-1000}$ in Graham et al [14].
(XLSX)

**S2 Table. Conditional analysis of LDL cholesterol-associated *LDLR* variants.** LDL cholesterol associations derived from Graham et al [14] for individuals of European ancestry before and after adjustment for linkage disequilibrium with the lead SNP (rs73015024). All noncoding variants within 500 kb of the *LDLR* TSS with an allele frequency greater than 1% were analyzed; those with significant associations after conditional analysis are displayed.
(XLSX)

**S3 Table. CRISPR library composition.** Identifier and sequences for every gRNA in the CRISPR library used in this study. For gRNAs targeting potential *LDLR cis*-regulatory sequences, the genomic coordinate for its predicted double-strand break and the basis for its selection are also provided.
(XLSX)

**S4 Table. CRISPR screen results.** DESeq2 output for the relative change in abundance of each indicated gRNA in LDL^high versus LDL^low cells. For each gRNA, the log-transformed median p-value for all gRNAs targeting the 100 bp window centered on its target site is also provided, together with the significance threshold for that window as determined by the number of gRNAs included in the window.
(XLSX)

**S5 Table. Genomic coordinates of *LDLR* CRE peaks.** Boundaries of the merged CRE peaks derived from merged rolling 100 bp windows demonstrating a significant reduction in LDL uptake upon their disruption.
(XLSX)

**S6 Table. MPRA library design.** Genomic coordinates and sequences for each indicated tile in the MPRA library. The test sequence constitutes the major allele at each position as determined by population allele frequences in Gnomad, notably including the presence of the common insertion rs59281581 that is not present in the hg38 human genome reference sequence.
(XLSX)

**S7 Table. Association of 5' UTR barcodes with *cis*-regulatory sequences.** The number of reads for each combination of 5' UTR barcode sequence and CRS test sequence are listed together with the overall share of reads for a given 5' UTR barcode that match the indicated CRS. Filtering was performed to remove any combination of 5' UTR barcode and CRS with less than 10 reads, or any 5' UTR barcode with less than 99% of reads associated with the same CRS.
(XLSX)

**S8 Table. MPRA expression data.** Aggregate counts per million for the 5' UTR barcodes corresponding to each *cis*-regulatory sequence in the MPRA library, demultiplexed by sample identity.
(XLSX)

**S9 Table. MPRA expression analysis.** Expression associated with each CRS, as quantified by the ratio of aggregate counts for its associated 5'UTR barcodes in cDNA relative to genomic DNA, for each of 5 independent replicates in HuH7 and HepG2 cells. Ratios are normalized to the average values for negative control sequences for each sample.
(XLSX)

**S10 Table. Oligonucleotides.** Sequences for the primers used in this study are provided and grouped by application.
(XLSX)

**S11 Table. Source data for *LDLR* candidate CRE nomination.** Identifying information for the ENCODE data sets used in the design of the custom CRISPR library for this study.
(XLSX)

## Author contributions

**Conceptualization:** Candilianne Serrano-Zayas, Brian T. Emmer.

**Formal analysis:** Kyle Leix, Candilianne Serrano-Zayas, Hitarthi S. Vyas, Sarah E. Graham, Brian T. Emmer.

**Investigation:** Kyle Leix, Candilianne Serrano-Zayas, Sarah E. Graham, Brian T. Emmer.

**Writing – original draft:** Kyle Leix, Candilianne Serrano-Zayas, Brian T. Emmer.

**Writing – review & editing:** Kyle Leix, Candilianne Serrano-Zayas, Hitarthi S. Vyas, Sarah E. Graham, Brian T. Emmer.

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
