## [Decision Letter · Decision Letter 0]

27 Nov 2025

PGENETICS-D-25-00962

Functional interrogation of candidate *cis* -regulatory elements at the *LDLR* locus

PLOS Genetics

Dear Dr. Emmer,

Thank you for submitting your manuscript to PLOS Genetics. After careful consideration, we feel that it has merit but does not fully meet PLOS Genetics's publication criteria as it currently stands. Therefore, we invite you to submit a revised version of the manuscript that addresses the points raised during the review process.

We look forward to receiving your revised manuscript.

Kind regards,

Takashi Fukaya

Academic Editor

PLOS Genetics

Pablo Wappner

Section Editor

PLOS Genetics

Aimée Dudley

Editor-in-Chief

PLOS Genetics

Anne Goriely

Editor-in-Chief

PLOS Genetics

**Journal Requirements:**

1) Please provide an Author Summary. This should appear in your manuscript between the Abstract (if applicable) and the Introduction, and should be 150-200 words long. The aim should be to make your findings accessible to a wide audience that includes both scientists and non-scientists. Sample summaries can be found on our website under Submission Guidelines:

https://journals.plos.org/plosgenetics/s/submission-guidelines#loc-parts-of-a-submission

2) We notice that your supplementary figures are uploaded with the file type 'Figure'. Please amend the file type to 'Supporting Information'. Please ensure that each Supporting Information file has a legend listed in the manuscript after the references list.

3) We notice that your supplementary Figures are included in the manuscript file. Please remove them and upload them with the file type 'Supporting Information'. Please ensure that each Supporting Information file has a legend listed in the manuscript after the references list.

4) Please amend your detailed Financial Disclosure statement. This is published with the article. It must therefore be completed in full sentences and contain the exact wording you wish to be published.

State what role the funders took in the study. If the funders had no role in your study, please state: "The funders had no role in study design, data collection and analysis, decision to publish, or preparation of the manuscript.".

**Reviewers' comments:**

Reviewer's Responses to Questions

**Comments to the Authors:**

Reviewer #1: In the manuscript entitled “Functional interrogation of candidate cis-regulatory elements at the LDLR locus”, Leix et al. investigated the LDLR gene locus, which is known to be associated with familial hypercholesterolemia and atherosclerotic cardiovascular diseases (ASCVD), to identify gene regulatory elements using CRISPR screening and massively parallel reporter assays (MPRA). They successfully identified a novel functional element located within the first intron of the gene. While they previously applied a similar CRISPR-based screening to analyze coding regions involved in LDL uptake, the present study extended the analysis to non-coding regions. The manuscript is well written; however, it has certain limitations primarily due to the methodology, and their findings remain somewhat limited in scope. Therefore, a more thorough discussion is considered necessary.

Major comments

1. Enhancers consist of multiple transcription factor binding sites, and their functional units typically span several hundred bps to about 1 kb. It is generally difficult to disrupt enhancer activity by introducing small-scale indels or point mutations using conventional CRISPR approaches. Therefore, enhancer analyses often require alternative methods such as ScanDel (which enables kilobase-scale deletions using paired sgRNAs; PMID: 28712454) or CRISPR interference/activation (CRISPRi/a) (PMID: 25307932, PMID: 31988385, PMID: 39294132), rather than standard CRISPR editing. Given this methodological limitation, the results presented in this study appear somewhat restricted (see further comments below). Although the authors briefly mention the limitations of conventional CRISPR methods (lines 273–276), they should provide a more detailed explanation about their methodological constraints with citing ScanDel and CRISPRi/a.

2. The regulatory activity of the LDLR promoter region and its variants has already been demonstrated in several previous studies using luciferase assays and MPRA (PMID: 11792717, PMID: 21538688, PMID: 31395865). In contrast to these studies, the CRISPR screening in this study did not detect any regulatory activity upstream of the gene. The authors should discuss this discrepancy, citing the above references. This is likely because, as mentioned earlier, the conventional CRISPR method may not have sufficiently disrupted the enhancer function. Conversely, I suspect that the strong activity observed downstream of the first exon was not caused by indels/mutations by the CRISPR, but rather by physical interference by the Cas9 complex (i.e., inhibition of transcription initiation and/or elongation).

3. The authors showed that two individual gRNAs reduced LDL uptake and LDLR mRNA levels in HuH7 cells (Fig. 7). However, these results were obtained from a heterogeneous cell population containing various indels. I would suggest that the authors isolate clonal cell lines, determine the exact sequence changes, and examine the resulting phenotypes.

4. The MPRA data were not described in sufficient detail. How many barcodes were associated with each tile? How was the correlation between experimental replicates?

5. MPRA and epigenetic data (ATAC-seq and H3K27ac peaks) identified the regions corresponding to tiles 6, 24, and 42 as the enhancer core (Figs. 5D, S6A). In contrast, CRISPR screening showed regulatory activity extending across the entire surrounding region (tiles 1–74) beyond the core (Fig. 5A). Please explain this discrepancy. Again, in my opinion, this may be due to interference with enhancer–TF or enhancer–promoter interactions caused by the Cas9 complex, rather than an actual disruption of enhancer sequence by CRISPR.

Reviewer #2: This manuscript, “Functional interrogation of candidate cis-regulatory elements at the LDLR locus,” presents a comprehensive and rigorous analysis of the cis-regulatory architecture governing LDLR expression. The authors combine GWAS information, open-chromatin and histone modification profiles, and transcription factor motif predictions to nominate candidate enhancer regions, followed by a well-executed CRISPR screening strategy that identifies functional elements influencing LDLR expression.

Importantly, the authors extend beyond screening alone: targeted deletion experiments convincingly validate enhancer activity within intron 1, and the use of massively parallel reporter assays (MPRA) provides high-resolution identification of core functional sequences. This multi-layered approach significantly strengthens the reliability of their conclusions.

The manuscript is clearly written, technically robust, and conceptually sound. The authors appropriately discuss the strengths and limitations of their approach, and the findings represent a valuable contribution to the fields of enhancer biology and gene regulation.

Reviewer #3: In this manuscript, Leix and co-authors conduct a CRISPR screen to identify novel enhancers of the LDLR gene and then examine in detail one of the newly-identified regions. The locus examined is of particular importance, as the expression levels of LDLR are tied to the likelihood of cardiovascular disease. The authors also make a compelling case that there were additional cis-regulatory elements to be uncovered, particularly those that are independent of the sterol response elements. In general, the paper is also well written and easy to follow and includes extensive and helpful supplementary information. My major concern has to do with the choice of CRISPR assay, which I detail below:

1. The author compile a large list of potential regulatory regions using GWAS hits, other variants of note, regions of open chromatin, regions with H2K27ac, regions of high conservation, and predicted cis-regulatory sequences in the vicinity of the LDLR TSS, and then test these using a CRISPR screen, which should generate InDels. I was surprised by the use of CRISPR vs. CRISPRi to identify regulatory regions. While CRISPRi has its own caveats, the authors point out that InDels near intron-exon boundaries may be false positives. But at the same time, I would expect many false negatives from bona fide regulatory regions that were not disrupted by a small InDel (this comes up in the discussion on line 279, but I think it should be addressed earlier and perhaps in more depth). The authors point out that some enhancer perturbations may be masked by the activity other enhancers in the locus, but there is often a high level of binding site redundancy within enhancers, making them notoriously robust to small mutations. I would be curious to at least see a discussion about the choice to use CRISPR vs. CRISPRi, if the approaches have been compared before for the identification of enhancers, and what the expected length distribution for the InDels generated with this approach would be. This might also be addressed by including positive control gRNAs that target known LDLR cis-regulatory regions, in addition those used already (which targeted coding regions).

2. Am I reading Figure 2 correctly in that all the CRE calls (except for one) lie within introns of the LDLR gene? I know intronic enhancers are not uncommon, but this was striking, especially since many of the tested gRNAs lay outside the gene itself, and the authors comment that none of the many regions containing upstream variants produced a signal. Why might that be? (As a quick reference, I found this paper, which estimated intronic enhancers as between ~40-75% of all enhancers, depending on tissue/developmental stage: https://pmc.ncbi.nlm.nih.gov/articles/PMC8327915/)

3. I’m not sure I totally agree with the claim on line 157 that the “conserved enhancer in the first LDLR intron… operates independently of SREBP binding” based on the evidence presented. Though the SRE motifs are >100 bps away, an enhancer can be as long as ~1000 bps, so those motifs might still influence the activity of that enhancer.

Minor concerns:

1. In the first paragraph of the results, can you briefly mention which tissues were used for the ATAC-seq, DHS-seq, and ChIP-seq analysis?

2. A zoom-in inset panel of Figure 2 that focuses on the region with all the CRE calls might be helpful.

3. A brief introduction to the HepG2 cells and why the lipoprotein-depleted state is needed for them in contrast to the HuH7 cells would be helpful on line 167.

4. Could some sort of scale bar be added to Figure 4 to indicate the size of the cloned LDLR^int1 region?

5. Figure 4B – I think there’s a typo in the y-axis label.

6. Which ones of the transcription factors named starting on line 298 are expressed in the cell lines used?

**Have all data underlying the figures and results presented in the manuscript been provided?**

Reviewer #1: Yes

Reviewer #2: Yes

Reviewer #3: Yes

PLOS authors have the option to publish the peer review history of their article (what does this mean? ). If published, this will include your full peer review and any attached files.

**Do you want your identity to be public for this peer review?** For information about this choice, including consent withdrawal, please see our Privacy Policy .

Reviewer #1: No

Reviewer #2: **Yes:** TATSUYA TAKEMOTO

Reviewer #3: No

**Figure resubmission:**
---

## [Decision Letter · Decision Letter 1]

22 Jan 2026

PGENETICS-D-25-00962R1

Functional interrogation of candidate *cis* -regulatory elements at the *LDLR* locus

PLOS Genetics

Dear Dr. Emmer,

Thank you for submitting your manuscript to PLOS Genetics. We invite you to submit a revised version of the manuscript that addresses the points raised by the Reviewer#3.

Please submit your revised manuscript within by Feb 21 2026 11:59PM. If you will need more time than this to complete your revisions, please reply to this message or contact the journal office at plosgenetics@plos.org. Please include the following items when submitting your revised manuscript:

We look forward to receiving your revised manuscript.

Kind regards,

Takashi Fukaya

Academic Editor

PLOS Genetics

Pablo Wappner

Section Editor

PLOS Genetics

Aimée Dudley

Editor-in-Chief

PLOS Genetics

Anne Goriely

Editor-in-Chief

PLOS Genetics

**Journal Requirements:**

**Reviewers' comments:**

Reviewer's Responses to Questions

**Comments to the Authors:**

Reviewer #1: The authors have addressed all of my comments. They have added sufficiently detailed discussion and experiments I requested.

Reviewer #3: I thank the authors for their detailed reply to my original review. The additional discussion and edits have improved the paper. I have one lingering comment: In my original comment about the use of CRISPR vs. CRISPRi, I asked about a positive control gRNAs that target known LDLR cis-regulatory regions. Was such a control done?

**Have all data underlying the figures and results presented in the manuscript been provided?**

Reviewer #1: Yes

Reviewer #3: Yes

PLOS authors have the option to publish the peer review history of their article (what does this mean? ). If published, this will include your full peer review and any attached files.

**Do you want your identity to be public for this peer review?** For information about this choice, including consent withdrawal, please see our Privacy Policy .

Reviewer #1: No

Reviewer #3: No

**Figure resubmission:**
---

## [Decision Letter · Decision Letter 2]

4 Mar 2026

Dear Dr Emmer,

We are pleased to inform you that your manuscript entitled "Functional interrogation of candidate *cis* -regulatory elements at the *LDLR* locus" has been editorially accepted for publication in PLOS Genetics. Congratulations!

Yours sincerely,

Takashi Fukaya

Academic Editor

PLOS Genetics

Pablo Wappner

Section Editor

PLOS Genetics

Aimée Dudley

Editor-in-Chief

PLOS Genetics

Anne Goriely

Editor-in-Chief

PLOS Genetics

BlueSky: @plos.bsky.social

Comments from the reviewers (if applicable):

Reviewer's Responses to Questions

**Comments to the Authors:**

Reviewer #3: The authors addressed my last question sufficiently. Thank you.

**Have all data underlying the figures and results presented in the manuscript been provided?**

Reviewer #3: None

PLOS authors have the option to publish the peer review history of their article (what does this mean? ). If published, this will include your full peer review and any attached files.

**Do you want your identity to be public for this peer review?** For information about this choice, including consent withdrawal, please see our Privacy Policy .

Reviewer #3: No

**Data Deposition**

http://datadryad.org/submit?journalID=pgenetics&manu=PGENETICS-D-25-00962R2

**Press Queries**

---

## [Editor Report · Acceptance letter]

PGENETICS-D-25-00962R2

Functional interrogation of candidate *cis* -regulatory elements at the *LDLR* locus

Dear Dr Emmer,

We are pleased to inform you that your manuscript entitled "Functional interrogation of candidate *cis* -regulatory elements at the *LDLR* locus" has been formally accepted for publication in PLOS Genetics! Your manuscript is now with our production department and you will be notified of the publication date in due course.

With kind regards,

Anita Estes

PLOS Genetics

On behalf of:
